# Using the Transformer Model for Physical Simulation: An application on Transient Thermal Analysis for 3d Printing Process Simulation

**Qian Chen**

Department of Mechanical Engineering and Materials Science
University of Pittsburgh
Siemens Digital Industries Software
qic35@pitt.edu

**Luyang(Chris) Kong**

Department of Applied Mathematics University of Washington
luyangkong@gmail.com

**Florian Dugast**

Department of Mechanical Engineering and Materials Science
University of Pittsburgh
fld5@pitt.edu

**Albert C. To**

Department of Mechanical Engineering and Materials Science
University of Pittsburgh
albertto@pitt.edu

## Abstract

Transient thermal analysis is widely used in many science and engineering areas such electronic package design, engine design and manufacturing. High-dimensional simulations are very expensive to run. Here we propose a machine learning model consisting of a pre-trained convolutional neural network (CNN), a transformer encoder and a multilayer perceptron (MLP) to predict the temperature field of 3D printed part. We use the low convolution layers of ResNet34 to extract low level geometry features from the CAD file, and transformer encoder to capture the long-range dependencies between layer-wise geometry features. The MLP then takes the transformer output as input and predicts the temperatures at any given locations and time. The results show the model can accurately predict the thermal history in 3D printing process on different geometries. Our model is also very efficient, runing 1~2 orders of magnitude faster than the simulation on which it is trained, without requiring the complicated pre-processing steps in transient thermal analysis including material property steup, mesh generation and refinement, and defining the boundary conditions and dynamic loading in every time step.

## 1 Introduction

Transient thermal analysis is essential in various engineering and scientific fields because it shows how the temperature distribution evolves over time in structures and systems as it plays an important

NeurIPS 2023 AI for Science Workshop.

role in their performance. Transient thermal analysis has been widely used in industries such as aerospace, automobile and consumer electronics. For example, in CPU and GPU design, it helps in predicting and mitigating the overheating issues that lead to component failure. However, high accuracy simulation results often requires fine time and spatial discretization which could be extremely computationally expensive. For example, simulations for 3D printing process aiming to capture the temperature rise and drop in millisecond scale could take days or even weeks.

In recent years, machine learning-based approaches have been introduced in finite element modeling (FEM) to approximate the expensive FEM solutions [1,2]. These surrogate models can provide rapid predictions of system behavior in scenarios such as engineering optimization where the finite element analysis (FEA) need to be performed many times with different design parameter combinations.In this paper, a machine learning model consisting of convolutional neural network (CNN), a transformer, and a multilayer perceptron is proposed for the transient thermal analysis of metal 3D printing process. Unlike the conventional subtractive manufacturing, metal 3D printing technique uses high-power laser or electron beam to fuse every layer of powder particles selectively and form a near-net-shape part in a layer-by-layer manner. Due to its capability to produce complex geometries with unique mechanical properties, this technique has gained increasing attention and has been widely used. Modeling the temperature evolution through the manufacturing process is very important to avoid build failures and ensure build quality. However, transient thermal simulations for 3D printing are usually expensive and time-consuming for part-scale model because of the incompatible spatial scale between laser or electron beam ($\mu m$) and the part (cm) to be simulated.

Graph neural networks have been applied to solve a variety of challenging and complex physics with mesh-based and particle-based simulators [3,4]. However, for high-dimensional and high-resolution transient simulations, the message-passing used to pass dynamics between mesh edges or particles in every time steps can be expensive under fine mesh configuration with thousands of time steps. As an emerging approach that seamlessly integrate data and physics, Physics-informed neural networks (PINNs) embed the partial differential equation (PDE) into the neural network as a part of the loss function [5]. Due to its versatility, PINNs have been applied to solve forward and inverse problems in fluid mechanics [6], solid mechanics [7] and heat transfer [8]. Even though hard constraints can be imposed to PINNs using penalty method and augmented Lagrangian method [9], PINNs are not a good choice for transient thermal simulation with changing boundary conditions and simulation domain. For example, in 3D printing, the part is built in a layer-by-layer manner. Once a new layer is formed, the simulation domain is modified while the new heat convection boundary should be applied on the newly formed layer. It would be very hard for PINNs to solve such problems. Recurrent neural network is designed for sequences of data, and is well-suited for transient thermal simulation tasks. Mozaffar et al [10, 11] proposed a recurrent neural network (RNN) structure with a Gated Recurrent Unit formulations for high-dimensional thermal history in 3D printing processes with variations in geometry, build dimensions, build strategy and laser parameters. This model could only performs on very simple geometry with very limited number of layers because of the RNN's incapability with long sequences.

In this paper, We propose a machine learning-based model for transient thermal simulation for 3D printing process at part level. The part geometry's effect on heat transfer through the whole process is considered by pre-trained convolution neural network and a transformer model. The remaining content of this paper is organized as follows. In Section 2, the similarities between transient thermal simulation and NLP tasks are discussed. In Sectoon 3, data prepartion and generation of the dataset for model training and evaluation are presented. In Section 4, the proposed model architecture is given. In Section 5, temperature profiles predicted by the proposed model are compared with ground truth.

## 2 Problem description

### 2.1 Governing equations of the simulation

The governing equation solved in transient thermal simulation for metal 3D printing process is:

$$\rho c \frac{\partial T}{\partial t} = \frac{\partial}{\partial x}(k\frac{\partial T}{\partial x}) + \frac{\partial}{\partial y}(k\frac{\partial T}{\partial y}) + \frac{\partial}{\partial z}(k\frac{\partial T}{\partial z}) + Q \tag{1}$$

where $T$ is the temperature, $t$ is the time, $\rho$ is the density of the material, $c$ is the heat capacity and $k$ is the thermal conductivity. $Q$ is the volumetric heat input for each layer. The elements for the layer to be built are activated and then the volumetric heat input is applied on this element layer. 3D-printed parts usually has a scale of centimeter and are divided into thousands of layers in the building. The real building takes a few hours or even days while its high-resolution simulation with thousands of time steps can take up to days to run.

## 2.2 Similarities between NLP task and transient thermal modeling

In this section, the intriguing parallels between Natural Language Processing (NLP) tasks and transient thermal simulation are explained. While seemingly distinct fields, both domains share a fundamental characteristics - the dynamic interplay of sequence vectors over time.

**Transient thermal modeling:** Transient thermal simulation is primarily concerned with comprehending how temperature evolves and fluctuates over time. At each discrete time step within the simulation, boundary conditions and dynamic loading parameters are meticulously defined before running the simulation. Importantly, the thermal profile established in preceding time steps have a significant influence on the temperatures encountered in the current step. This intricate interdependence between sequences of load and boundary conditions highlights the inherently temporal nature of transient thermal modeling. Moreover, similar to context in NLP, the geometry of the simulation domain also influences the temperature profiles in profound ways. The shape and dimensions of the domain dictate how the heat is conducted. For example, different geometric shapes can offer varying conduction pathways for heat transfer. This will be discussed in detail in Section 2.3.

**NLP tasks:** In NLP tasks such as sentimental analysis and language translation, the sequence and context of words within a given text carry substantial importance. Much like the transient thermal system, the order in which words are encountered in a sentence, paragraph, or document can drastically alter the meaning and interpretation. Context, built upon previous words and phrases, plays a pivotal role in discerning the intent and sentiment behind language. The parallel becomes even more evident when We consider the dynamic nature of NLP tasks. Just as transient thermal modeling dynamically adjusts boundary conditions and loading at each time step, NLP models adapt to changing contexts as they process language. Each word encountered is akin to a step in time, with the embedded vector representations capturing the linguistic nuances of the past, shaping the understanding of the present, and influencing the predictions of the future.

In essence, the varying boundary conditions and dynamic loading in each time step of transient thermal modeling find a curious reflection in the embedded vectors used in NLP. Both domains illustrate the profound importance of temporal context in interpreting and modeling complex systems. By recognizing these similarities, we gain valuable insights into how the principles of one field can inform and inspire advancements in the other, paving the way for interdisciplinary synergy and innovation.

## 2.3 Long-range dependencies in transient thermal modeling

In the 3D printing process, the part is built in a layer-by-layer fashion. Every layer is melted by a high-power laser beam and fused with the previous layers. This process is repeated many times to build the part from bottom to top, and usually takes days or weeks for part in centimeter scale. The previously deposited part serve as the cooling channels to dissipate heat from top layers to the build tray. For part 1 and part 2 in Figure 1(a), even though their top parts are identical, different temperature profiles of the top layers (layer #1000) are expected since they have different bottom geometries. The cooling channel for Part 1 is a solid block which is able to dissipate more heat than Part 2 the cooling channels of which are three thin columns. In transient thermal analysis, this effect on the temperature field can be easily captured since the finite element analysis solves the PDE step by step. High-resolution transient thermal simulations for the 3D printing process usually have thousands of time steps and take days or even weeks to run due to its high spatial and temporal scale. Machine learning models such as Recurrent Neural Network (RNN) and Long-short term memory (LSTM) [12] shown in Figure 1(b) could be used for transient thermal analysis. However, sequence to sequence (seq2seq) models are not able to handle such long sequences with over thousands of time steps. The transformer architecture [13] that was proposed in 2017 and revolutionized the NLP tasks relies on self-attention mechanism, and allows the model to capture dependencies and relationships

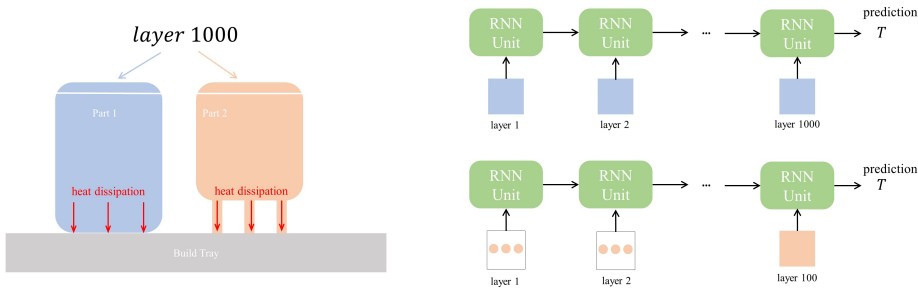

(a) Similar geometries with different base

(b) Simulation with RNN

Figure 1: Long-range dependencies in transient thermal simulation

between words in a sentence regardless of their position. We use the transformer in our model in hope that the model could learn the geometric change on heat conduction in 3D printing even though the layers that may affect the heat transfer are far in distance and have a very long time span.

## 2.4 Transformer encoder

Large language models can be categorized into encoder-only and decoder-only by its architecture. We only used the decoder of the transformer model for time profile prediction. Encoder-only large language models such as BERT [14] enable it to consider both left and right contexts when encoding tokens. Encoder-only models focus on encoding inputing sequences into rich contextual embeddings while decoder-only models such as GPT3 [15] only consider the left context window and are used to generate output sequence tokens autoregressively. The simulation task of interest needs the complete sequences that contains geometry, boundary conditions and thermal load for temperature prediction. Therefore, the transformer encoder is used in the proposed model.

## 3 Data collection

Unlike images for computer vision and text data for NLP, collecting simulation data is very challenging. Firstly, due to the high-dimensional and large-scale characteristics of the simulation data, the data collection process for scientific simulations is less cost-effective and readily accessible compared to gathering images or text from the Internet. Simulations may run for extended periods, limiting the speed at which data can be generated and imposing time constraints on collecting sufficient quantities of data for machine learning tasks. The simulation data could be overwhelming, requiring substantial computational resources and storage capacity. Furthermore, engineering simulations are often domain-specific. For example, the collected stress data cannot be used to train a model for transient thermal simulator. Moreover, simulations frequently encompass a wide range of parameters and boundary conditions, leading to an exponentially growing dataset space that is difficult to explore comprehensively.

As high-quality public data is not readily available, we have employed our proprietary GPU-based FEM simulation software [16] to execute simulations on various geometries and acquire the necessary dataset for training our model. It is impractical to sample every node temperature from the simulation, given that the node count typically reaches millions, and the transient thermal simulation for 3D printing processes entails thousands of time steps. For each geometry, the sliced layer-wise cross sections are taken as 224 x 224 gray scale images. we adopted two distinct data sampling approaches as shown in Figure 2: First, we sampled specific nodes and tracked their temperature variations over time, with the intention of enabling the model to learn the temporal evolution of temperature profiles. Secondly, at selected time intervals, we comprehensively sampled the temperature of every node within the simulation, aiming to facilitate the model's grasp of spatial temperature distributions at specific time points. For each node, the collected data pair format is $<$ (x,y,z,time) | temperature$>$. The first element in the pair will be mapped to higher dimensional space and concatenate with the geometry vector while the second element is the ground truth.

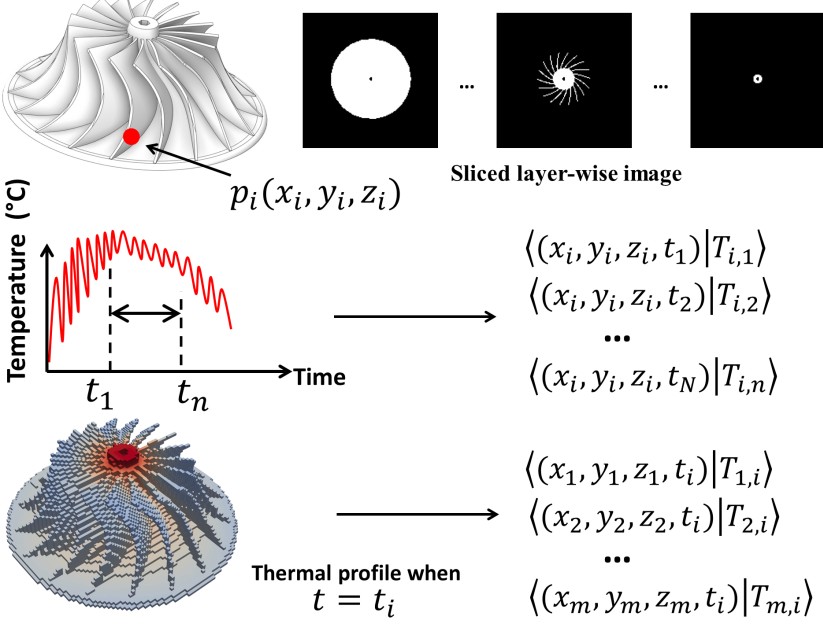

Figure 2: Collect data from simulation

## 4 Model

### 4.1 Model architecture

In contrast to finite element simulation which discretizes the geometry into elements, our model utilizes sliced layer-wise images as input for geometry processing. Rather than processing a vast number of elements, this approach is cost-effective and highly efficient while also accounting for the geometry influence on heat transfer. As shown in Figure 3, We firstly process each sliced layer-wise image with the pre-trained ResNet34 [17], and extract the feature map from its conv4 layers for low-level geometry features. We apply a max-pool operation on the feature map and obtain a 2048 x 1 embedding vector for each layer. We adopt the positional encoding in neural radiance fields (NeRF) [18] to map the 4 x 1 query vector $(x, y, z, time)$ to a higher dimensional space to better fit the high frequency temperature variation (See the thermal history in Fig. 2). The encoding function is:

$$\gamma(p) = (sin(2^0\pi p), cos(2^0\pi p), ..., sin(2^{L-1}\pi p), cos(2^{L-1}\pi p)) \tag{2}$$

Where $L$ is frequency number. We use $L = 32$ in this model, and map the 4 x 1 query vector into 256 x 1, and concatenate with the layer-wise geometry embedding vector. The transformer processes the layer-wise embedding vectors and the multilayer perceptron takes the averaged output from transformer, and predict the temperature at $(x, y, z, time)$.

### 4.2 Model training

We collected a dataset which contains the pairs $< $ (x,y,z,time) | temperature$>$ for a variety of geometries and the geometry's layer-wise images. 640,000 pairs were used to train the model on a machine with 4 NVIDIA A100 GPUs. The model training process has 50 epochs which takes 48 hours. We used the AdamW optimizer and applied dropout for regularization.

## 5 Results

We test the performance of the proposed model on different geometries, and compared the results to FEM-based simulation ground truth. We also calculate the accuracy of temperature predictions. For each node, if the relative difference between predicted temperature and ground truth falls within the threshold $\tau = 5 \times 10^{-2}$, the prediction is considered as correct; Otherwise, it is regarded as incorrect.

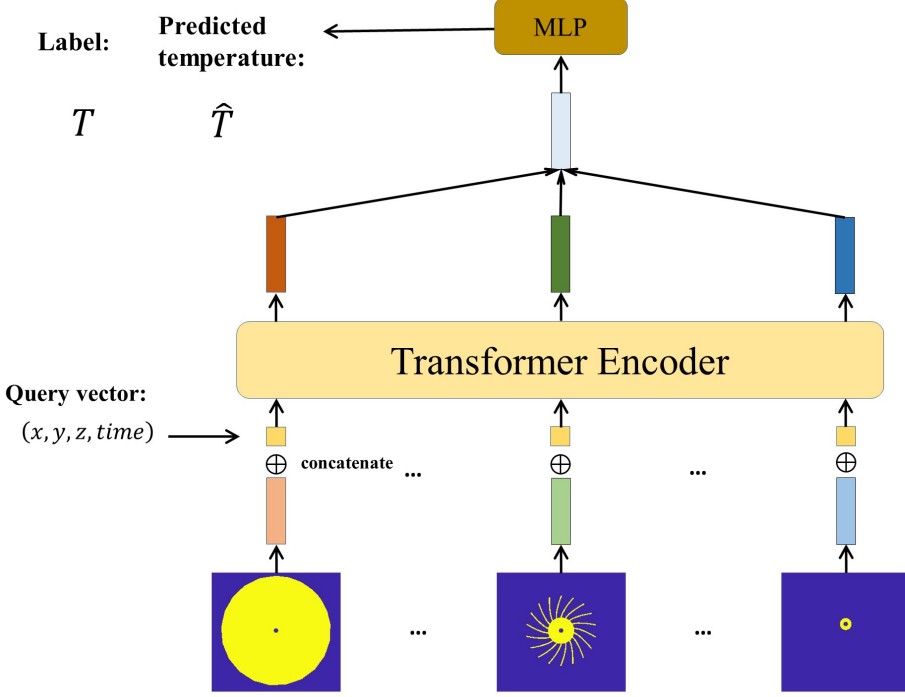

Figure 3: Model architecture

Our main findings are that the proposed model is able to produce high-quality temperature profile at unseen time steps, while being significantly faster than the GPU-based ground truth simulation software, and simplify the simulation workflow.

While our model boasts a parameter count three orders of magnitude larger than those utilizing RNN and LSTM units, the comparison of performance metrics, RMSE and accuracy, highlights our model's superiority over its counterparts (see Table 1). In Figure 5, the predicted profile of the complex geometries at steps unseen at training stage could match the ground truth with acceptable variance. The spatial temperature distribution is reasonable while the magnitude difference is acceptable.

Table 1: The transformer model achieves better performance than models with RNN and LSTM

| Models | # Parameters | RMSE ($\times 10^{-3}$) | Accuracy | |
|---|---|---|---|---|
| | | | $\tau$ | $2\tau$ |
| CNN + RNN(base) + MLP | 378,113 | 103.34 | 0.09 | 0.18 |
| CNN + RNN(large) + MLP | 1,051,649 | 82.94 | 0.14 | 0.36 |
| CNN + LSTM(base) + MLP | 1,312,769 | 71.55 | 0.41 | 0.53 |
| CNN + LSTM(large) + MLP | 3,414,017 | 59.08 | 0.51 | 0.63 |
| CNN + Transformer + MLP | 113,335,297 | 9.48 | 0.79 | 0.91 |

Table 2 presents the simulation time by FEM-based simulation software and the inference time by our proposed model on different geometries. The time step number of transient thermal simulation for 3D printing depends on the part's layer number which makes the FEM-based simulation extremely time-consuming. A distinguishing feature of our model is its ability to predict temperature at any given step without prior knowledge of the temperature profile from preceding steps. The inference time for any step is at the scale of seconds. For the temperature evolution of the entire part, our model exhibits a prediction speed approximately 10~15 times faster.

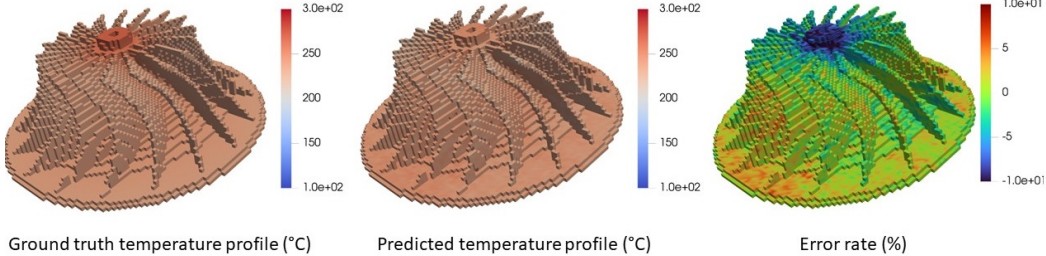

Ground truth temperature profile (°C)   Predicted temperature profile (°C)   Error rate (%)

Figure 4: The temperature profile for the compressor case

Table 2: The performance of our model on different geometries

| Geometry | # layers | # time steps | Simulation time | Inference time |
|---|---|---|---|---|
| Compressor | 750 | 15,000 | 1 hour 15 mins | 4 mins |
| Turbine blade | 2,574 | 51,480 | 2 hours 8 mins | 11 mins |
| Air Bracket | 1,550 | 31,000 | 6 hours 10 mins | 32 mins |

## 6 Conclusion

We presented a data-driven model for transient thermal simulation with CNN, the transformer model, and multilayer perceptron. Because of the similarities between NLP and transient thermal modeling, the decoder of transformer is employed to process the layer-wise embedding vectors of the 3D printed part. The model We proposed has outstanding performance on the dataset collected from high-resolution simulations. The model has a inference time 1∼2 orders of magnitude faster than the FEM-based simulation running on GPU. We are planning to make the code and dataset open in the public domain.

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
