# OpenReview forum: "Using the Transformer Model for Physical Simulation: An application on Transient Thermal Analysis for 3D Printing Process Simulation"
_NeurIPS.cc/2023/Workshop/AI4Science — NeurIPS2023-AI4Science Poster_

### Official Review · Reviewer_7pRP · 2023-10-15
**Good application of sequential models to an important engineering problem, but results need some work**

**Rating:** 6
**Confidence:** 5

**Review:**

This work compares different NN architectures for an important engineering task tied to 3D printing.

The authors show that transformers outperform other models.

I do have an issue with the conclusions of the paper, where the authors state that Transformers can outperform other models. However, this may be due to the number of parameters in their transformer model which greatly exceed the other shown models.

The authors might want to review neural scaling laws [1] as concept. To summarize bigger models are expected to be more accurate. Thus further analysis is needed to show either small transformer models or large lstm (and other) models to draw a valid conclusion from this.

However, for the purposes of a workshop. The demonstrated work is sufficient. Please take into account my comments if you're pushing this for a future publication.

1. Hoffmann, Jordan, Sebastian Borgeaud, Arthur Mensch, Elena Buchatskaya, Trevor Cai, Eliza Rutherford, Diego de Las Casas et al. "Training compute-optimal large language models." arXiv preprint arXiv:2203.15556 (2022).

---

### Official Review · Reviewer_TCCd · 2023-10-24
**Transforming 3D Thermal Simulations: A Few Gaps to Address**

**Rating:** 7
**Confidence:** 4

**Review:**

This manuscript introduces a method to create a surrogate model for transient thermal simulations of 3D printing processes. The objective of the surrogate model is to condense computer-intensive calculations, which can take weeks, into mere minutes. The proposed machine learning architecture is transformer-based, serving as an autoregressive predictor for temperature in time series data. This approach aims to circumvent the need for costly FEM simulations of heat PDEs. Although 640,000 data points (x,y,z,time) were collected, and the model was trained using corresponding temperature labels, the manuscript lacks details on the geometries of these data points. There is also ambiguity regarding generalization across different geometries. Another point of concern is that while RMSE values are provided, the typical temperature for such systems (e.g., 200°C?) is absent. Without this context, it's challenging to discern if there's a meaningful difference between RMSE values ranging from 0.1 to 0.009. If such differences aren't practically significant, the transformer model might not offer advantages over a CNN + RNN(base) + MLP architecture, prompting questions about the choice of transformers. Despite these shortcomings, I recommend this paper for acceptance.

---

### Meta-Review · Area_Chair_SNnS · 2023-10-27

**Recommendation:** Accept (Poster)
**Confidence:** 4

**Metareview:**

The paper presents a method to create a surrogate model for transient thermal simulations of 3D printing processes using transformer-based machine learning architecture. Reviewers express concerns about missing details regarding the geometries of the data points and the generalization across different geometries. They also highlight the absence of context for RMSE values. Moreover, reviewers question the conclusion about transformers outperforming other models and suggests considering neural scaling laws for a valid comparison. However, they find the work suitable for a workshop publication.